# A Deep Learning Approach for Prognostic Evaluation of Lung Adenocarcinoma Based on Cuproptosis-Related Genes

**DOI:** 10.3390/biomedicines11051479

**Published:** 2023-05-19

**Authors:** Pengchen Liang, Jianguo Chen, Lei Yao, Zezhou Hao, Qing Chang

**Affiliations:** 1Shanghai Key Laboratory of Gastric Neoplasms, Department of Surgery, Shanghai Institute of Digestive Surgery, Ruijin Hospital, Shanghai Jiao Tong University School of Medicine, Shanghai 200020, China; liangpengchen@shu.edu.cn; 2School of Microelectronics, Shanghai University, Shanghai 201800, China; 3School of Software Engineering, Sun Yat-sen University, Zhuhai 528478, China; 4School of Health Science and Engineering, University of Shanghai for Science and Technology, Shanghai 200093, China

**Keywords:** lung adenocarcinoma, cuproptosis-associated genes, deep neural network, individualized prognostic models

## Abstract

Lung adenocarcinoma represents a significant global health challenge. Despite advances in diagnosis and treatment, the prognosis remains poor for many patients. In this study, we aimed to identify cuproptosis-related genes and to develop a deep neural network model to predict the prognosis of lung adenocarcinoma. We screened differentially expressed genes from The Cancer Genome Atlas data through differential analysis of cuproptosis-related genes. We then used this information to establish a prognostic model using a deep neural network, which we validated using data from the Gene Expression Omnibus. Our deep neural network model incorporated nine cuproptosis-related genes and achieved an area under the curve of 0.732 in the training set and 0.646 in the validation set. The model effectively distinguished between distinct risk groups, as evidenced by significant differences in survival curves (*p* < 0.001), and demonstrated significant independence as a standalone prognostic predictor (*p* < 0.001). Functional analysis revealed differences in cellular pathways, the immune microenvironment, and tumor mutation burden between the risk groups. Furthermore, our model provided personalized survival probability predictions with a concordance index of 0.795 and identified the drug candidate BMS-754807 as a potentially sensitive treatment option for lung adenocarcinoma. In summary, we presented a deep neural network prognostic model for lung adenocarcinoma, based on nine cuproptosis-related genes, which offers independent prognostic capabilities. This model can be used for personalized predictions of patient survival and the identification of potential therapeutic agents for lung adenocarcinoma, which may ultimately improve patient outcomes.

## 1. Introduction

Lung adenocarcinoma (LUAD) is a significant threat to public health and a major contributor to cancer-related fatalities, accounting for the majority of lung cancer cases [1,2,3,4]. Despite recent advancements in treatment, the five-year survival rate for LUAD patients remains low, hovering around 15% [5]. The increasing incidence and mortality rates of LUAD [6,7] highlight the pressing need for effective and accurate diagnostic and prognostic tools. In recent years, the development of novel prognostic models has emerged as a promising approach to improving the understanding of tumor progression and predicting patient survival. These models are often based on various molecular biomarkers and clinical traits, aiming to provide more precise prognostic information and guide personalized treatment strategies. Recent studies have reported a range of prognostic models for lung adenocarcinoma. These include models based on endoplasmic reticulum stress-related gene scores [8], 7-methylguanosine (m7G) related gene signatures [9], methylation and CD8 T cell signatures [10], mRNA and lncRNA associated with prognosis and immunity [11], and models based on differentially expressed genes and clinical data using Lasso penalty and cross-validation methods [12]. Additionally, models based on metabolic genes [13], RNA-binding proteins (RBPs) [10], and N6-methyladenosine (m6A)-related long noncoding RNA (lncRNA) [14] have also been developed. Taken together, these prognostic models offer a diverse array of tools for clinicians to better understand the development of lung adenocarcinoma, enhance treatment efficacy, and provide more accurate prognostic information for patients.

Early detection and accurate prognostic tools are essential for managing LUAD, but current biomarker prognostic models require further development. Prognostic models based on various molecular biomarkers and clinical traits have emerged as promising approaches to improving the understanding of tumor progression and predicting patient survival. Meanwhile, copper, an essential element in human health, has gained increasing attention due to its role in cancer development and cell death [15]. Copper is a cofactor for essential enzymes, and normal cells maintain low levels of copper through a homeostatic mechanism to prevent the accumulation of harmful intracellular free copper [16]. Recently, a novel form of cell death known as cuprotosis has been discovered, which is triggered by the accumulation of copper in cells and results in proteotoxic stress caused by mitochondrial respiration [17]. Elevated copper levels have been detected in both the serum and tumor tissue of cancer patients [18]. This has led to the exploration of copper ion carrier-related drugs and copper chelators as potential antitumor therapies.

In this study, we analyzed the differential expression of 13 cuproptosis-associated genes (CRGs) provided by TSVETKOV et al. [17] in LUAD data. Based on the differential expression of these CRGs, we constructed a deep neural network-based prognostic model of LUAD risk and compared it with the COX risk model. The model was evaluated for both its ability to make independent predictions and its ability to make individualized predictions. The gene expression profiles of patients who were identified as high-risk or low-risk by the model were analyzed and differences were found. These differences in gene expression were then used to predict drugs that may be effective in treating LUAD.

## 2. Materials and Methods

### 2.1. Datasets

We utilized sequencing and clinical data of LUAD from the TCGA database (version 33.0). The TCGA database provided sequencing data from 555 samples, obtained from 486 patients, including 501 tumor samples and 54 normal samples. The clinical data consisted of age, gender, stage, T status (tumor size and extent), N status (lymph node involvement), and M status (distant metastasis) for each patient. For external validation, we also utilized sequencing and clinical data from the Gene Expression Omnibus (GEO) database (GSE68465 [19]). The GEO database provided sequencing data from 443 samples, obtained from 443 patients, all of which were tumor samples. The accompanying clinical data consisted of age, gender, stage, T status (tumor size and extent), N status (lymph node involvement), and M status (distant metastasis) for each patient. In addition, we utilized progression-free survival (PFS) data from the TCGA database and tumor mutational burden (TMB) data from the Cancer Genome Atlas (https://portal.gdc.cancer.gov/projects, accessed on 1 October 2022). We adopted the cuproptosis-related genes (CRGs) identified by Tsvetkov et al. in their study [17], which were determined based on a comprehensive review of previous literature. The CRGs include *ATP7B, ATP7A, PDHB, PDHA1, DLAT, DLST, GCSH, DBT, DLD, LIAS, LIPT1, SLC31A1*, and *FDX1*.

The flowchart of our study is illustrated in Figure 1. We started by collecting data from The Cancer Genome Atlas (TCGA) and the Gene Expression Omnibus (GEO) databases. Following data collection, we conducted a differential expression analysis to screen for differentially expressed genes. Based on these results, we constructed a prognostic model using Deep Neural Network (DNN) techniques. This model was then validated using independent data from the GEO database. Lastly, we performed functional analysis and evaluated the performance of our model, focusing on its ability to predict the prognosis of lung adenocarcinoma (LUAD) based on cuproptosis-related genes (CRGs).

### 2.2. Differentially Expressed Gene Screening and DNN Model Construction

Gene selection and normalization. The Wilcoxon test was applied for difference analysis and nine CRGs were selected as differentially expressed genes (DEGs) with a *p*-value of <0.05. The data from TCGA and GEO were normalized to a range of 0–1 to account for sequencing errors between the two datasets.

In our study, we constructed a Deep Neural Network (DNN) model for prognostic prediction in lung adenocarcinoma (LUAD) using the differentially expressed genes (DEGs) obtained from The Cancer Genome Atlas (TCGA) data. The choice of a DNN model was driven by its superior ability to capture complex, non-linear relationships between predictors and the response variable; in this case, patient survival time. This capability becomes especially relevant when handling high-dimensional genomic data where intricate interactions are expected. On the contrary, while the Cox proportional hazards model has been widely utilized in survival analysis due to its interpretability and statistical properties, it assumes a linear relationship between the log-hazard and the predictors. Further, the proportional hazards assumption of the Cox model may not fully encapsulate the complex relationships inherent in genomic data, leading to potential compromises in predictive accuracy. Our DNN model (as shown in Figure 2) comprises one input layer, three hidden layers, and one output layer. This architecture was strategically chosen to balance model complexity and computational efficiency. To prevent model overfitting, we employed a hybrid regularization approach, combining L1 and L2 regularization methods. The L1 regularization promotes feature sparsity, aiding in the identification of the most significant features, while the L2 regularization curbs model complexity by penalizing large weights. For hyperparameter optimization, we utilized Bayesian optimal tuning methods. The optimal hyperparameters, which offered the best model performance during tuning, included a learning rate of 0.2, a learning rate decay of 0.9999, a tanh activation function, L1 regularization of 0.0009382036085065045, L2 regularization of 0.000007640570141805595, and the Adam optimization function. To validate our choice of the DNN model, we compared its performance with the Cox model using the area under the curve (AUC) of the receiver operating characteristic (ROC) at different time points.

Model Evaluation. The performance of the DNN model was evaluated using the C-index curve and the loss curve. The final output of the model was the patient’s risk score, calculated as the mean squared error (ME) defined as: ME = $\frac{1}{n}\sum_{i=1}^{n}{\left(y_{i}-\hat{y_{i}}\right)}^{2}$, where *n* is the number of patients with observable events and $\hat{y_{i}}$ is the output of the network.

### 2.3. Based on the Determination of Model Grouping and Model Rationality Analysis

The performance of the DNN model was evaluated and compared with the conventional Cox prognostic model using receiver operating characteristic (ROC) curves at 1, 3, and 5 years. The best cut-off value for patient risk stratification was determined using the Youden index based on the 1 year ROC curve. This threshold was chosen as it maximized the sensitivity and specificity of the model. The differences in survival between the subgroups were subsequently analyzed using the Kaplan–Meier (KM) survival curve. Lastly, the independent prognostic ability of the DNN model was assessed using both univariate and multivariate Cox models.

### 2.4. Progression-Free Survival (PFS) Analysis and Clinical Characteristics Exploration

The PFS data were combined to evaluate the significance of the differences in PFS by subgroup. A heatmap was generated to visualize the relationship between the subgroups and various clinical characteristics. Furthermore, the clinical characteristics were grouped into two stages (early I–II and late III–IV) and the relationship between the subgroups and these stages was explored.

### 2.5. Functional Analysis of Differences between Model Subgroups

To investigate the differences between model subgroups at the genetic, immune, and tumor mutation levels, separate functional analyses were performed. Firstly, the differences between the subgroups in gene pathways were analyzed through KEGG and GO enrichment analyses. Subsequently, the immune microenvironment was assessed using an ssGSEA approach, which analyzed 16 immune cells and 13 pathway activities. The ‘maftools’ package was employed for Tumor Mutational Burden (TMB) analysis, which included calculating the mutation load score and the number of mutated genes per sample. Samples were then divided into high or low TMB groups based on a threshold that maximized the difference in survival curves (according to the log-rank test from the Kaplan–Meier survival analysis). The top 15 genes were analyzed in each group. Lastly, the combination of TMB and DNN was used to analyze overall survival.

### 2.6. Development of Nomogram Model for Individualized Clinical Decision Making

To account for the individualized differences between patients, we established an individualized clinical decision-making nomogram model. The model was constructed by selecting DNN, stage, T, and N based on the analysis of the C-index of DNN and clinical characteristics at 0–10 years. The model was visualized using a high-risk patient and a low-risk patient as examples. The rationality of the model was evaluated using the C-index and its reliability was further assessed at 1, 3, and 5 years using calibration curves.

### 2.7. Screening of Anti-Tumor Sensitive Drugs

To identify potential drugs for the treatment of high-risk LUAD patients, we utilized the “pRRophetic” package to analyze the differences in sensitivity of 251 drugs between the different risk groups. This was based on their IC50 values, with a lower IC50 value indicating increased sensitivity. Significance was determined if the *p*-value fell within the established range. The aim was to identify drugs with greater sensitivity in the high-risk group.

### 2.8. Effect of Sensitive Drugs on the Activity of LUAD A549 Cell Line

This study aimed to evaluate the impact of BMS-754807 on the A549 cell line derived from lung adenocarcinoma (LUAD). The evaluation was performed using two assays, cell proliferation and cell migration. For the cell proliferation assay, log-growing A549 cells were plated in 96-well plates and treated with various concentrations of BMS-754807 (0, 0.01, 0.1, 0.5, 1, 10, and 20 μM) for 24 and 72 h. Cell viability was then assessed using the CCK-8 assay. The cell migration assay involved treating A549 cells with different concentrations (0, 0.01, and 0.1 μM) of BMS-754807 for 48 h, followed by collection, counting, and inoculation into the upper chamber of a transwell system. The lower chamber was filled with complete medium containing 10% FBS, and cells were allowed to migrate for 12 h. After migration, the cells were fixed, stained, and analyzed under an inverted microscope.

### 2.9. Statistical Analysis

In this study, we implemented a comprehensive statistical and machine learning analysis to ensure the robustness and validity of our findings. The statistical analyses, deep learning model construction, and graphical representations were performed using R (version 4.0.4) and RStudio (version 1.4.1103) software, which are widely recognized for their versatility and reliability in data analysis and machine learning tasks. We utilized a Deep Neural Network (DNN) to construct a prognostic model for lung adenocarcinoma. DNNs are a class of artificial neural networks that excel in learning complex patterns and relationships from high-dimensional data. They are known for their ability to model non-linear and intricate relationships, making them suitable for our study where multiple factors are interacting in complex ways. To evaluate the independence of factors in our model, we set a stringent significance level of *p* < 0.001 for both univariate and multivariate Cox models. This strict criterion helped minimize the chances of false positives and provided more confidence in the independence of the factors identified by our DNN model. It also indicated that our model is capable of predicting patient outcomes independently of other known clinical and pathological factors. In the process of screening for sensitive drugs, we used a significance level of *p* < 0.05. This less stringent criterion allowed us to identify potential therapeutic agents that might have a significant impact on patient outcomes, while still controlling for the possibility of false discoveries.

## 3. Results

### 3.1. Identifying and Modeling Cuproptosis-Related Differentially Expressed Genes

Our objective was to investigate the association between CRGs and patient prognosis in LUAD. Utilizing the TCGA dataset, we performed a differential gene expression analysis, which allowed us to identify nine differentially expressed genes (DEGs) associated with cuproptosis (Figure 3a). The TCGA dataset was processed by excluding 25 samples that lacked survival status features, and normalizing the remaining 461 samples to a 0–1 scale. Subsequently, we constructed a DNN prognostic prediction model (Figure 3b,c) and assessed its accuracy by comparing it with the traditional Cox model. The comparison revealed the superior performance of the DNN model in terms of the area under the ROC curve (AUC) at different time points: Year 1 (0.732 vs. 0.646), Year 3 (0.777 vs. 0.637), and Year 5 (0.836 vs. 0.594) (Figure 3d,e). To stratify the patient samples into high-risk and low-risk groups, we determined the optimal threshold value by calculating the Youden index (value of 0.8758, Figure 3f). The resulting survival curves displayed a statistically significant difference in survival times between the high- and low-risk groups (p<0.001, Figure 3g). We further validated the independence of our model through analysis of variance (Figure 3h,i). The findings underscore the robust predictive capability of the DNN model developed in this study, emphasizing its potential for generating personalized predictions of patient survival probabilities in the context of LUAD.

### 3.2. External Validation Using GEO Dataset

To validate the performance of our DNN model, we utilized the GEO dataset comprising 443 selected samples. These samples were first normalized to a “0–1” scale and were subsequently subjected to risk score calculation using the DNN model. The samples were then stratified into distinct risk groups based on the optimal threshold value derived from the Youden index. The DNN model’s performance was evaluated by plotting ROC curves (Figure 4a) for 1, 3, and 5 years and then comparing it with the conventional Cox model (Figure 4b). The comparison demonstrated the superiority of the DNN model over the Cox model at all three time points (Year 1: 0.606 vs. 0.601; Year 3: 0.621 vs. 0.586; Year 5: 0.603 vs. 0.584), indicating the robustness and improved predictive ability of our model (as shown in Table 1). Additionally, we generated survival curves (Figure 4c) for the validation set, which revealed a considerable divergence between the risk groups. The high-risk group exhibited a shorter time to half-death compared to the low-risk group, emphasizing the model’s capacity to discriminate between patients with different prognoses. To further verify the model’s independence and generalizability, we conducted an analysis of variance (Figure 4d,e). The results confirmed that the DNN model maintained its predictive power and independence in the external validation set, highlighting its potential as a reliable prognostic tool for clinical applications.

### 3.3. Association between Model-Based Risk Stratification and Clinical Characteristics

We investigated the association between the risk group stratification generated by our model and various clinical characteristics of the patients. The Kaplan–Meier survival curves depicted in Figure 5a showcase distinct differences in progression-free survival (PFS) among the identified model subgroups. This outcome highlights the model’s capacity to effectively discriminate patient survival based on the assigned risk groups. To gain a deeper understanding of the relationship between the model subgroups and specific clinical characteristics, we focused on the variables T, N, and stage, as they are well-known for their significant influence on the prognosis of LUAD patients. The heatmaps presented in Figure 5b visually represent these associations, emphasizing the considerable disparities between subgroups concerning these clinical variables. Subsequent statistical analyses confirmed the significant effects of T (p<0.001), N (p<0.001), and stage (p<0.001) on the risk stratification. To further demonstrate the model’s effectiveness in differentiating patient outcomes within specific clinical contexts, we divided the patients into early-stage (I and II) and late-stage (III and IV) subgroups. Separate survival curves were plotted for each of these groups (Figure 5c,d), revealing the model’s ability to stratify patients across different clinical stages while maintaining statistically significant distinctions in survival (p<0.001). These findings underscore the model’s potential to serve as a valuable tool in guiding clinical decision-making by providing more nuanced insights into patients’ prognoses based on their individual clinical characteristics.

### 3.4. Functional Analysis of Model Grouping

To delve deeper into the biological significance of the different risk groups, we performed Gene Ontology (GO) and Kyoto Encyclopedia of Genes and Genomes (KEGG) functional enrichment analyses. These analyses were intended to identify potential differences in gene function and metabolic pathways among the risk groups. The differentially expressed genes (DEGs) were identified using a Wilcoxon test, with a false discovery rate (FDR) of <0.05 and an absolute log2 fold change (|log2FC|) of ≥1 serving as the selection criteria. We identified 514 DEGs, of which 508 were down-regulated and only six were up-regulated. Enrichment analyses of these DEGs pointed to significant variations in functions such as striated muscle cell differentiation, sarcomere organization, and immune receptor activity, as visualized in Figure 6a. Discrepancies were also observed in pathways, particularly in the metabolism of xenobiotics by cytochrome P450 (Figure 6b). Furthermore, we noticed distinct differences in the immune microenvironment between risk groups. We performed a single sample gene set enrichment analysis (ssGSEA) to evaluate 16 immune cell enrichments and 13 pathway activities, finding that the high-risk group exhibited lower levels of immune cell and pathway activities than the low-risk group. This difference was especially pronounced for Tfh, aDCs, B cells, Mast cells, Neutrophils, Th1 cells, and TIL as shown in Figure 6c,d. To explore the potential impact of tumor mutational burden (TMB) on patient survival, we calculated TMB scores for each sample and categorized them into high or low TMB groups. Waterfall plots of the top 15 genes in both TMB groups highlighted patterns of difference between them (Figure 6e,f). Finally, we integrated the TMB analysis with our risk model and found that patients with a high-risk score and low TMB had the poorest survival outcomes, thereby suggesting that our risk model might be a stronger predictor of survival than TMB alone (Figure 6g).

### 3.5. Construction of a Nomogram Model for Personalized Clinical Decision-Making

In order to account for the inherent heterogeneity among patients and further enhance the utility of our findings in clinical scenarios, we proceeded to develop a nomogram model. This model was designed to provide clinicians with a practical tool to predict individual patient outcomes based on the integration of our DNN model and select clinical features. Our initial step involved the comparison of the C-index across a span of 0 to 10 years for the DNN and various clinical characteristics (Figure 7a). In this analysis, we identified that the DNN, along with the clinical characteristics of stage, T, and N, exhibited superior performance with C-index values exceeding 0.6, indicating a satisfactory degree of discrimination. Given their predictive prowess, these four elements were selected to construct our nomogram model. To illustrate the practical application of this model, we presented two representative patient scenarios: one high-risk patient (TCGA-NJ-A7XG, Figure 7b) and one low-risk patient (TCGA-78-7540, Figure 7c). The overall performance of the nomogram was quantified by the C-index, which reached 0.795, showcasing a robust ability of the model to predict patient outcomes accurately. Finally, to validate the predictive accuracy and reliability of our model, calibration curves were plotted for 1, 3, and 5 years (Figure 7d). The close alignment of the calibration plots with the 45-degree line served as a testament to the reliability of our model, underlining its potential for effective individualized prognostic prediction in clinical settings.

### 3.6. Anti-Tumor Susceptibility Drug Screening and Sensitivity Results

We screened 251 antitumor drugs to identify potential therapeutic agents for lung adenocarcinoma. Based on the screening results, we found that BMS-754807 (p=0.011) exhibited significant sensitivity as a potential drug for treatment (Figure 8). Figure 8a shows the screening of 251 anti-tumor drugs, revealing that BMS-754807 was highly sensitive, and Figure 8b presents the structural diagram of the BMS-754807 compound. To further investigate the potential therapeutic effect of BMS-754807, we conducted a series of experiments. The CCK-8 assay results revealed that BMS-754807 inhibited the proliferation of LUAD A549 cell lines at various concentrations (0, 0.01, 0.1, 0.5, 1, 10, and 20 μM) and durations (24 h, Figure 9a, and 72 h, Figure 9b). The proliferation effect decreased significantly with increasing BMS-754807 concentration (*p* < 0.01). Furthermore, cell migration results (Figure 9c,d) showed a significant reduction in the number of migrated cells when exposed to BMS-754807 at concentrations of 0.01 μM and 0.1 μM. The lowest number of migrated cells was observed at a concentration of 0.1 μM. These results indicate that various concentrations of BMS-754807 can inhibit the migration of LUAD A549 cells, with higher concentrations resulting in a stronger inhibition effect.

## 4. Discussion

Lung cancer, specifically LUAD, continues to be a significant cause of mortality worldwide, with a high incidence and prevalence [20,21]. The late-stage diagnosis, lymph node involvement, and multiple metastases commonly seen in LUAD patients contribute to the poor prognosis [22]. This underscores the need for continued research in identifying novel biomarkers and developing prognostic models to improve patient outcomes. Our study leveraged the advances in next-generation sequencing techniques to explore the role of 13 cuproptosis-related genes (CRGs) in LUAD. We conducted a Wilcoxon test analysis on lung adenocarcinoma gene expression data from TCGA, from which we identified nine differentially expressed CRGs including *ATP7B, PDHA1, PDHB, LIPT1, LIAS, FDX1, SLC31A1, DLD*, and *DLAT*. These genes, implicated in various biological processes and disease states, were used to construct a prognostic risk model.

In this model, ATP7B, a copper transporter, has been shown to contribute to platinum drug resistance in various cancer cells [23,24,25,26]. The genes PDHA1 and PDHB, both part of the pyruvate dehydrogenase complex, are reported to influence oxidative phosphorylation, tumor growth, metastasis, and glycolysis regulation [27,28,29,30,31]. Other genes such as *LIPT1, FDX1*, and *SLC31A1* also play important roles in tumor growth and patient prognosis [17,32,33,34,35,36,37,38,39,40,41,42]. The implications of these findings extend beyond improving our understanding of the molecular mechanisms of LUAD. These genes could be potential targets for therapeutic interventions, and their expression patterns could serve as prognostic biomarkers in clinical settings.

In recent years, the development of gene-based prognostic models and their applications in immunotherapy have gained widespread attention. Wang et al. [43] constructed a survival risk prediction model based on the expression of four m6A-related genes, suggesting their potential as diagnostic and prognostic factors. Li et al. [44] identified differentially expressed m6A RNA methylation regulators in lung adenocarcinoma and used them to construct a risk signature, showing a strong association with clinical outcomes and prognosis. Immune cell infiltration (ICI) in the tumor microenvironment (TME) offers insights into the prognosis of immunotherapy. One study aimed to create an ICI scoring model and evaluate its ability to predict the effects and prognosis of immunotherapy for lung adenocarcinoma patients. The results indicated that this scoring system accurately predicted overall survival for these patients [45]. Another study employed non-negative matrix factorization (NMF) to develop a model based on immunogenic cell death-related genes, which assessed the survival prognosis of lung adenocarcinoma. This NMF model provided valuable guidance for lung adenocarcinoma prognosis [46].

Cuprotosis, a novel and distinct form of cell death, has significant implications in various cancers. Xiaona et al. [47] investigated the role of CRGs in the tumor microenvironment (TME) of LUAD, using data from The Cancer Genome Atlas (TCGA) and Gene Expression Omnibus (GEO) databases. They analyzed the connections between various subgroups, clinical pathological traits, and immune infiltration features with the TME mutation status. Their study aimed to enhance the clinical application of CRG scores and estimate the survival probability of patients. Jiang et al. [48] explored the potential relationship between cuproptosis-related anoikis prognostic genes (ANRGs) and clinicopathological features, TME, and mutation characteristics in LUAD. They constructed a risk score model incorporating seven ANRGs signatures and developed a highly reliable nomogram to help formulate treatment strategies based on risk score and the clinicopathological features of LUAD. In comparison, our study utilized a DNN model to predict LUAD prognosis based on nine CRGs. Our model effectively distinguished between distinct risk groups and provided personalized survival probability predictions. Furthermore, our study identified the drug candidate BMS-754807 as a potentially sensitive treatment option for LUAD. Our study presented a unique approach to predicting LUAD prognosis by employing a DNN model based on CRGs. While Xiaona et al. [47] and Jiang et al. [48] explored the roles of CRGs in different aspects of LUAD, our study contributes to the field by offering a novel prognostic model that can be used for personalized predictions of patient survival and the identification of potential therapeutic agents for LUAD, ultimately aiming to improve patient outcomes.

## 5. Conclusions

To conclude, our study highlights the potential of a DNN prognostic model based on cuproptosis-related genes in predicting patient outcomes in LUAD. We identified several crucial genes involved in tumor growth, copper metabolism, and patient prognosis, which can potentially serve as targets for therapeutic interventions. Furthermore, our DNN model demonstrates robust independent predictive capabilities, making it a valuable tool for personalized risk assessment and treatment planning. The potential of BMS-754807 in inhibiting LUAD cell growth and migration emphasizes the importance of further research into the drug’s efficacy and clinical applications. Collectively, our findings contribute to the growing body of knowledge on LUAD, supporting the development of novel therapeutic strategies and prognostic tools to ultimately improve patient outcomes.

## Figures and Tables

**Figure 1 biomedicines-11-01479-f001:**
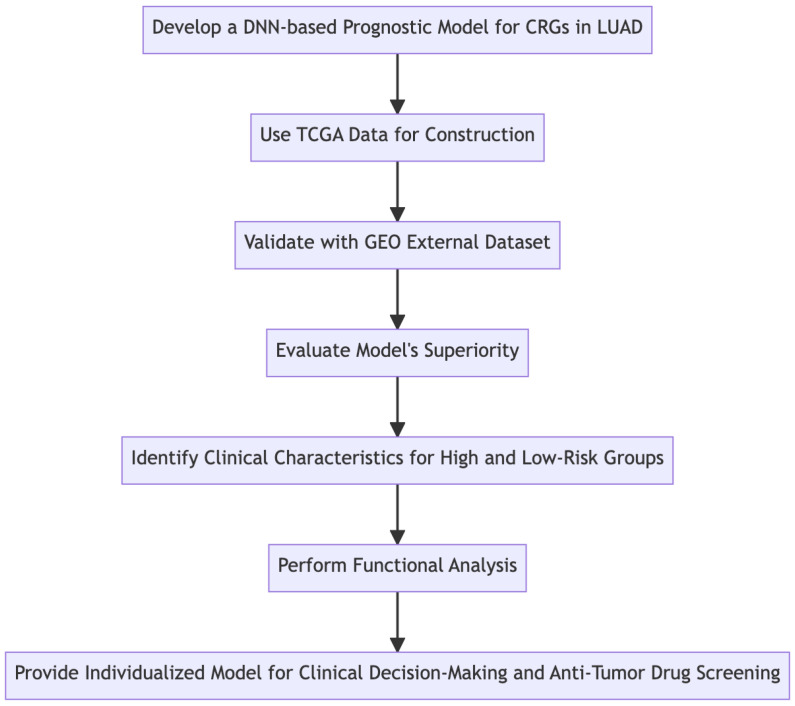
Construction and evaluation of a DNN-based prognostic model for CRGs in LUAD utilizing TCGA data. The flowchart presents the study process, starting with data collection from TCGA and GEO databases, followed by differential expression analysis, model construction using DNN, model validation using GEO data, and finally functional analysis and assessment of the model’s performance.

**Figure 2 biomedicines-11-01479-f002:**
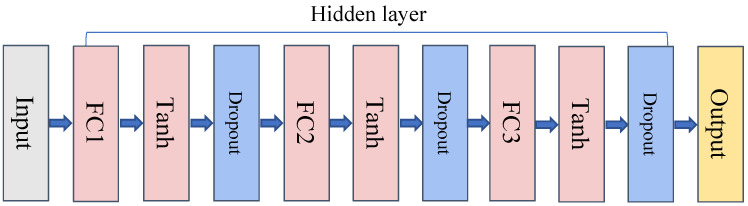
Schematic representation of the Deep Neural Network (DNN) model used in this study. The model consists of an input layer, three hidden layers, and an output layer. Fully Connected Layer (FC) that connects each neuron to all neurons in the previous layer. Activation Function (tanh) is applied after the fully connected layer, introducing non-linearity into the model. Dropout is then applied to prevent overfitting by randomly dropping out a fraction of input neurons. The Output Layer generates risk scores for each patient. Although not explicitly shown in the diagram, all weights in the model are subject to L1 and L2 regularization during training to further prevent overfitting and encourage feature sparsity.

**Figure 3 biomedicines-11-01479-f003:**
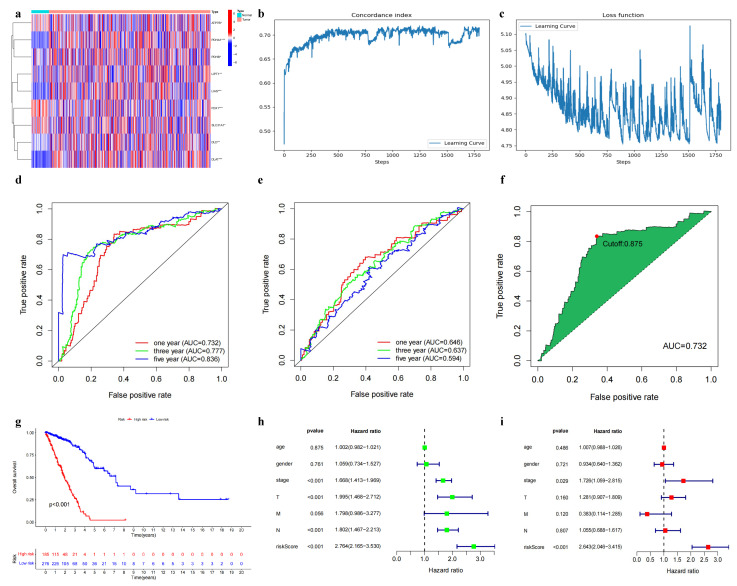
Screening cuproptosis-related differential genes using TCGA data and developing a DNN model. (**a**) Nine cuproptosis-related differential genes.Asterisks represent levels of statistical significance: * *p* < 0.05, ** *p* < 0.01, *** *p* < 0.001. (**b**) C-index plot of the DNN prognostic model. (**c**) Loss plot of DNN prognostic model. (**d**) ROC curve of DNN prognostic model. (**e**) ROC curve of the Cox prognostic model. (**f**) Determining Youden index values (optimal thresholds for separating the different risk groups) through ROC curves.The red dot represents the optimal cutoff point. (**g**) Survival curves based on subgroups. (**h**) Single-factor Cox independence test. (**i**) Multi-factor Cox independence test.

**Figure 4 biomedicines-11-01479-f004:**
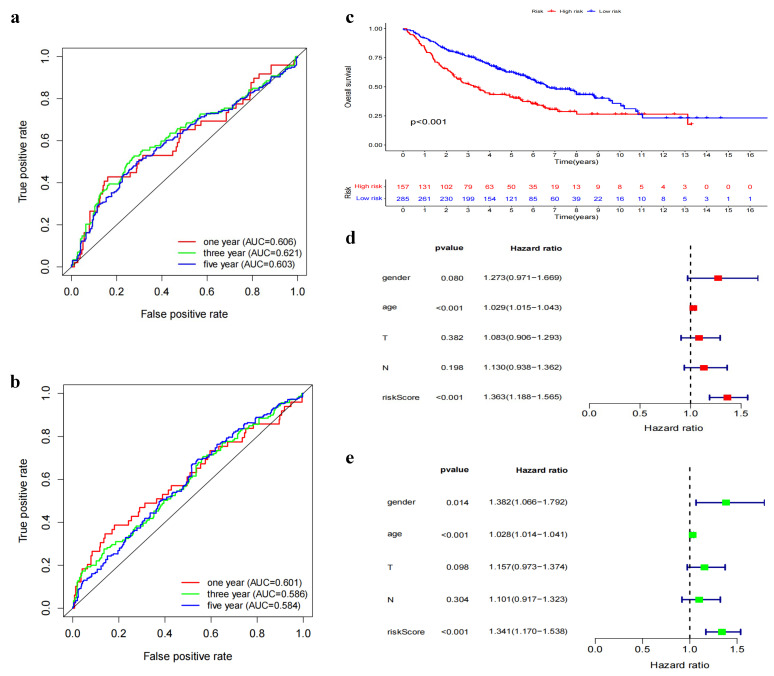
GEO external validation. (**a**) DNN prognostic model ROC curve. (**b**) cox prognostic model ROC curve. (**c**) Subgroup-based survival curves. (**d**) Single-factor Cox independence test. (**e**) Multi-factor Cox independence test.

**Figure 5 biomedicines-11-01479-f005:**
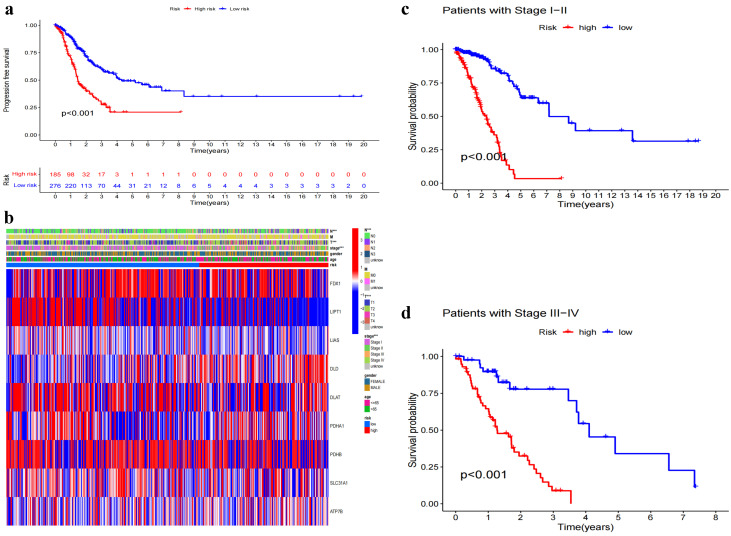
Differences in clinical characteristics between the different risk groups. (**a**) Survival curves based on progression-free survival between subgroups. (**b**) Heatmap based on differences in clinical characteristics (age, gender, stage, T,M,N) between subgroups, *** *p* < 0.001. (**c**) Survival curves based on early stage (I and II) patients between subgroups. (**d**) Survival curves based on late stage (III and IV) patients between subgroups.

**Figure 6 biomedicines-11-01479-f006:**
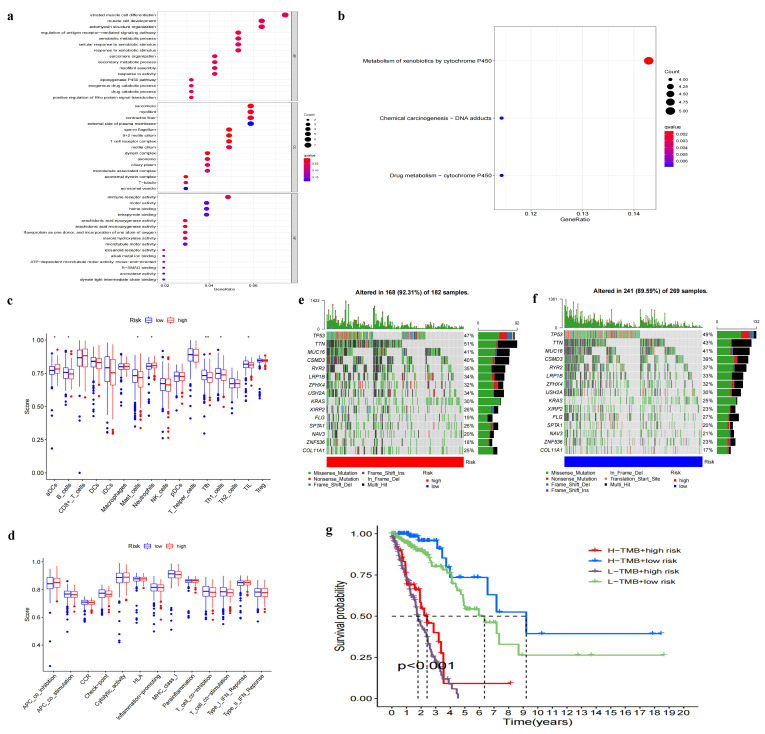
Functional differences between the different risk groups. (**a**) GO analysis. (**b**) KEGG analysis. (**c**) ssGSEA-based differential analysis of 16 immune cells, * *p* < 0.05, ** *p* < 0.01. (**d**) ssGSEA-based differential analysis of 13 immune pathways. (**e**) Waterfall plot of high mutation load patients based on the top fifteen most mutated genes. (**f**) Waterfall plot of low mutation load patients based on the top fifteen most mutated genes. (**g**) Survival curves based on the combination of high–low TMB and high–low risk.

**Figure 7 biomedicines-11-01479-f007:**
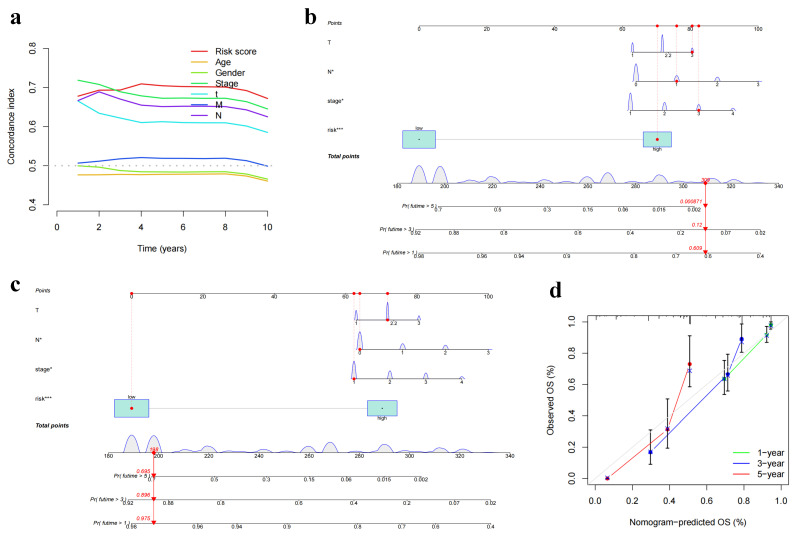
Individualized prognostic prediction nomogram model building, * *p* < 0.05, *** *p* < 0.001. (**a**) Plot of 0–10 year C-index curve based on DNN (riskscore) and clinical information. (**b**) Nomogram model of a high-risk patient TCGA-NJ-A7XG. (**c**) Nomogram model of a low-risk patient TCGA-78-7540. (**d**) Calibration curves for model years 1, 3, and 5.

**Figure 8 biomedicines-11-01479-f008:**
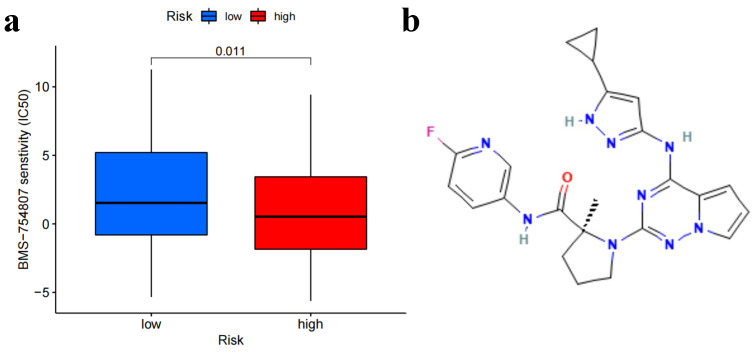
Anti-tumour sensitivity drug screening. (**a**) The screening of 251 anti-tumour drugs revealed that BMS-754807 was highly sensitive. (**b**) Structural diagram of BMS-754807 compound.

**Figure 9 biomedicines-11-01479-f009:**
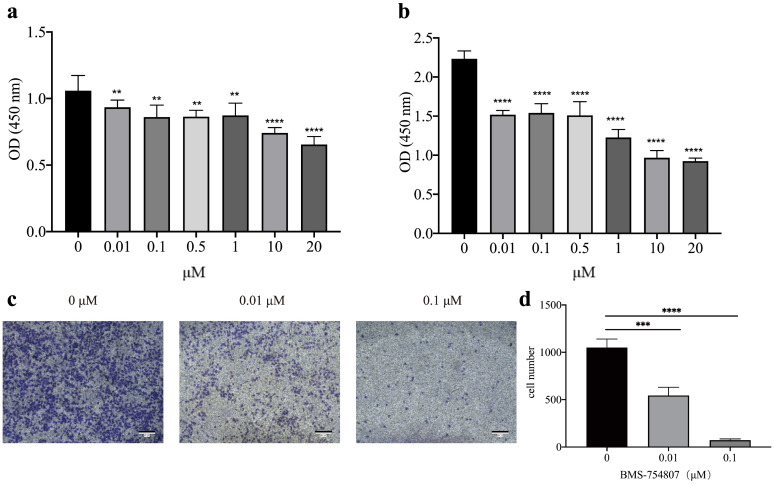
Effect of BMS-754807 on the activity of LUAD A549 cell line. (**a**) 24 h of CCK-8 experiment results, ** *p* < 0.01, **** *p* < 0.0001. (**b**) 72 h of CCK-8 experiment results, **** *p* < 0.0001. (**c**) The migration ability of different concentrations of BMS-754807 on A549 cells was assayed by Transwell chamber. (**d**) Quantitative statistics of migration ability, *** *p* < 0.001, **** *p* < 0.0001.

**Table 1 biomedicines-11-01479-t001:** Comparison of AUC between DNN and Cox model at different time points in the GEO validation dataset.

Model	Year 1 AUC	Year 3 AUC	Year 5 AUC
DNN	0.606	0.621	0.603
Cox	0.601	0.586	0.584

## Data Availability

The datasets utilized in this study were obtained from the TCGA database (https://portal.gdc.cancer.gov/projects/, accessed on 1 October 2022) and the GEO database (https://www.ncbi.nlm.nih.gov/geo/, accessed on 1 October 2022, accession ID: GSE68465).

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
