# Peer review of "A Deep Learning Approach for Prognostic Evaluation of Lung Adenocarcinoma Based on Cuproptosis-Related Genes"

_biomedicines, 2023, doi:10.3390/biomedicines11051479_

Round 1

Reviewer 1 Report

This is an apparently interesting article, but it requires significant revision for publication. Some aspects to consider are indicated below:

Line 55 indicates the clinical data used in the dataset. But it is not specified what exactly each of them are. Given the importance I think it would be interesting to include it.

In line 62 the CRGs used are indicated, but their choice is not justified.

The DNN model used is very basic and above all it is not explained why this model was chosen and not another one. I think that not only the model used and its parameters should be explained, but also why that model was chosen and why those parameters were used and not others. Perhaps a figure of the model where L1 and L2 are explained would be interesting.

The section "Based on the Determination of Model Grouping and Model Rationality Analysis" I think should be better explained, explaining the parameters and curves used and not only naming them, as well as referencing them adequately.

In general it is convenient that the figures are "close" to their references in the text, so as not to get lost.

The results are promising and well illustrated. Although a real discussion of them is needed in the next section.

The discussion section I think should be completely revised as there is an introduction that should be at the beginning of the article, as nothing is discussed. A central part that could remain as part of the discussion, although the discussion of the results obtained would really be missing. Leaving the final part for the beginning of the conclusions section that is completely missing, since the final analysis of the conclusions that can be obtained from the results is not carried out.

Author Response

Response to Reviewer 1 Comments

This is an apparently interesting article, but it requires significant revision for publication. Some aspects to consider are indicated below:

Point 1. Line 55 indicates the clinical data used in the dataset. But it is not specified what exactly each of them are. Given the importance I think it would be interesting to include it.
Response 1: Thank you for your suggestions. We have clarified in the revised manuscript the specific clinical data that were used, including patients' age, gender, staging, tumor size and extent (T status), lymph node involvement (N status), and distant metastasis (M status).

Point 2. In line 62 the CRGs used are indicated, but their choice is not justified.

Response 2: Thank you for your suggestions. We have included background information on Cuproptosis-Related Genes in the revised manuscript and mentioned the cuproptosis-associated genes identified by Tsvetkov et al. from previous literature.

Point 3. The DNN model used is very basic and above all it is not explained why this model was chosen and not another one. I think that not only the model used and its parameters should be explained, but also why that model was chosen and why those parameters were used and not others. Perhaps a figure of the model where L1 and L2 are explained would be interesting.

Response 3: Thank you for your valuable comments and suggestions. We have made corresponding modifications and improvements in the manuscript.

Firstly, we chose the DNN model because of its superior performance in capturing complex nonlinear patterns and relationships in high-dimensional data, which is particularly important for gene expression data. Compared to other models such as support vector machines (SVM) or random forests, DNN may perform better in this aspect. Meanwhile, as the Cox model is commonly used in prognosis model studies, we also conducted a detailed comparison of the performance between DNN and Cox models in the paper and found that DNN performed better.

Secondly, in terms of the selection of the model architecture, we used an input layer, three hidden layers, and an output layer. The selection of these layers was to strike a balance between model complexity and computational efficiency. At the same time, we applied L1 and L2 mixed regularization to prevent overfitting of the model. L1 regularization encourages feature sparsity, which helps identify the most important features, while L2 regularization controls model complexity by penalizing large weights. The combination of L1 and L2 regularization can provide a balance between sparsity and complexity in the model.

Finally, we used Bayesian optimization to determine the most suitable combination of hyperparameters. These hyperparameters generated the best model performance during the tuning process.

Thank you again for providing valuable feedback, which has been very helpful for my paper.

Point 4. The section "Based on the Determination of Model Grouping and Model Rationality Analysis" I think should be better explained, explaining the parameters and curves used and not only naming them, as well as referencing them adequately.

Response 4: We appreciate your feedback and agree with your insights, which have provided valuable input to our research. In our revised manuscript, we have made modifications to the "Functional analysis of model groups" section to provide more specific explanations and additional details. We not only explained the parameters and curves used, but also cited relevant studies to support our approach. These changes should convey our intentions and findings more clearly.

Point 5. In general it is convenient that the figures are "close" to their references in the text, so as not to get lost.

Response 5: Thank you for your valuable suggestion. We completely agree that placing figures and tables as close as possible to their references in the text can help readers better understand and follow our research. In our revised manuscript, we have adjusted the positions of the figures and their references to improve the readability and comprehensibility of the article. In addition, to further enhance readability, we will provide more context and transitions between the figures and their references in the text, which will help readers better understand the role of the figures in our study.

Point 6. The results are promising and well illustrated. Although a real discussion of them is needed in the next section. The discussion section I think should be completely revised as there is an introduction that should be at the beginning of the article, as nothing is discussed. A central part that could remain as part of the discussion, although the discussion of the results obtained would really be missing. Leaving the final part for the beginning of the conclusions section that is completely missing, since the final analysis of the conclusions that can be obtained from the results is not carried out.

Response 6: Thank you for your valuable feedback. Your suggestions have been very beneficial to our work. With regards to the revision of the Discussion section, we appreciate your feedback and fully agree that our discussion should delve more deeply into our research findings. In the revised Discussion section, we have provided a more in-depth discussion of our findings, particularly on how these findings impact our understanding of LUAD and how they can be translated into more effective treatment strategies and prognostic tools. We also discussed the independent predictive ability of our DNN model and its potential value in personalized risk assessment and treatment planning. Furthermore, we further discussed the ability of BMS-754807 to inhibit LUAD cell growth and migration and emphasized the importance of further research on its efficacy and clinical application. Finally, we explicitly pointed out in the Discussion section that our study has contributed to the research on LUAD and supported the development of new treatment strategies and prognostic tools to ultimately improve patients' prognosis.

Thank you very much for your careful review and constructive feedback on our article.

Reviewer 2 Report

The primary aim of the  research proposed by the authors was to develop a deep neural network (DNN) model for predicting the prognosis of lung adenocarcinoma (LUAD) by identifying cuproptosis-related genes (CRGs).

Data was obtained from The Cancer Genome Atlas (TCGA) and the Gene Expression Omnibus (GEO) databases. Initially, differentially expressed genes were screened from TCGA data through differential analysis of CRGs. Subsequently, a prognostic model was established using DNN and validated with GEO data. The performance of this model was compared to the traditional Cox model, and further assessed via gene enrichment analysis, immune microenvironment examination, and tumor mutation burden (TMB) functional analysis.

The DNN model incorporated 9 CRGs and displayed an AUC of 0.732 in the training set and 0.646 in the validation set.

The model effectively separated the data into distinct risk groups, exhibiting significant differences in survival curves (P<0.001) and demonstrating significant independence as a standalone prognostic predictor (P<0.001). Functional analysis revealed differences in cellular pathways, immune microenvironment, and TMB between the various risk groups. Moreover, the model offered personalized survival probability predictions with a C-index of 0.795, and the drug candidate BMS-754807 was identified as a potentially sensitive treatment option.

The authors concluded that the DNN prognostic model for LUAD, based on 9 CRGs, possesses independent prognostic capabilities and can be utilized for personalized predictions and the identification of potential therapeutic agents for LUAD patients.

Interesting article.

I suggest some minor revisions.

1)      The abstract must better summarize the sections. Now it directly starts with the aim.

2)      Add a clear and effective purpose.

3)      Figure 1 must be described in details.

4)      Check the resolution of the figures

5)      Avoid short paragraphs (see for example Anti-tumor Susceptibility Drug Screening)

6)      Check the MDP standard for the text. The references for example are listed without a heading…

Author Response

Response to Reviewer 2 Comments

Point 1. The abstract must better summarize the sections. Now it directly starts with the aim. Add a clear and effective purpose.

Response 1: Thank you very much for your valuable feedback. Your suggestions have been very helpful for us to deeply reflect on our paper and make further improvements. Following your suggestions, we have made significant revisions to the abstract to better summarize each section and clarify our research purpose.

Point 2. Figure 1 must be described in details.

Response 2: Thank you very much for your suggestions. Based on your feedback, we have made detailed modifications to the description of Figure 1 in the original manuscript.

Point 3. Check the resolution of the figures

Response 3: Thank you very much for your feedback. We have adjusted the resolution of the images in the manuscript according to your suggestion.

Point 4. Avoid short paragraphs (see for example Anti-tumor Susceptibility Drug Screening)

Response 4: Thank you for your suggestions. We have merged the two subsections of "Antitumor susceptibility drug screening" and "Sensitivity drug (BMS-754807) activity results" to enhance the coherence of the paragraph. We believe that these modifications will help readers better understand our work.

Point 5. Check the MDP standard for the text. The references for example are listed without a heading…

Response 5: Thank you for the reminder. We have carefully checked the manuscript to ensure that it conforms to the MDP format standard. Regarding the issue raised by the reviewer about the reference list lacking a title, we have added an appropriate heading "References". In the revised version, we have also made sure that all punctuation, citations, and formatting conform to the journal's requirements. We hope that these improvements meet the expectations of the reviewer and the editor, and we thank you once again for your constructive feedback, which has been very helpful to our work.

Reviewer 3 Report

In this manuscript the authors have developed a deep neural network model for predicting the prognosis of lung adenocarcinoma by identifying cuproptosis-related genes. Authors need to compare their studies with the state of the art works.

1. Avoid using the abbreviations/acronyms in the abstract.

2. The authors could expand the discussion on the statistical significance of the present study.

3. Could the authors comment about the confidence value used in this work and how was it determined.

4. The authors need to compare their findings with recent cuproptosis-related gene signatures reported in the literature.

https://www.sciencedirect.com/science/article/abs/pii/S0010482523002962

https://www.sciencedirect.com/science/article/pii/S2405844023012987

5. The authors will need to show more functional experiments to establish the role of genes in immune therapy and test its therapeutic potential.

6. Expand the logical validation of the experiments with a clear discussion.

7. Authors should improve the figures as they are unreadable, small, and congested. It is recommended to use vector graphics images. 

8. How did the author determine each type of immune cell in the database? 

9. How was the optimal thresholds determined? Clearly clarify this point wherever it has been used. 

10. The section and sub-section numbering is missing.

11. Captions/titles of the figures should be improved. For e.g. Figure 1 title can be made concise.

12. The legends, axes titles in the figures should be stated clearly. 

13. Include a separate section for the related work.

14. Include a separate section for the conclusions.

Moderate editing of English language

Author Response

Response to Reviewer 3 Comments

Point 1. In this manuscript the authors have developed a deep neural network model for predicting the prognosis of lung adenocarcinoma by identifying cuproptosis-related genes. Authors need to compare their studies with the state of the art works.

Response 1: Thank you for the valuable feedback. We have consulted relevant literature and compared our research with the latest related studies in the revised manuscript. We have emphasized the uniqueness of our model, particularly in the way we focus on Cuproptosis-Related Genes and the superiority of our deep neural network model in predicting the prognosis of lung adenocarcinoma. We believe that by integrating these features, our work makes a significant contribution to the study of prognosis prediction for lung adenocarcinoma. Once again, we appreciate your constructive feedback, which has been very helpful to our work.

Point 2. Avoid using the abbreviations/acronyms in the abstract.

Response 2: Thank you for your valuable feedback. We have carefully reviewed the abstract and revised the use of abbreviations and acronyms to make it more understandable. We greatly appreciate your guidance.

Point 3. The authors could expand the discussion on the statistical significance of the present study.

Response 3: Thank you for your valuable feedback. We have expanded the discussion on statistical significance in the statistical methods section of the paper.

Point 4. Could the authors comment about the confidence value used in this work and how was it determined.

Response 4: Thank you for your valuable suggestions, the following are my thoughts on confidence, please review: The confidence level used in a study is a measure of the reliability of the statistical results. It is a metric used to estimate the likelihood that the statistical result can be replicated in another identical study. The choice of the confidence level often depends on the nature of the study and the field of research. In many scientific fields, a 95% confidence level is commonly used, which suggests that the results would be replicated in 95 out of 100 identical studies.  In our study, the selection of the confidence level was based on the standard practices in the field of biomedical research. We used a 95% confidence interval (CI) for our survival probability predictions. The 95% CI is a range within which we would expect the true value to fall 95% of the time if the study were repeated multiple times under the same conditions.  However, it's important to note that the confidence level does not reflect the probability that the specific hypothesis being tested is true. Rather, it reflects our confidence in the methodology used to test the hypothesis.  In terms of how we determined our confidence level, as with many statistical decisions, this was based on a balance between precision and reliability. A higher confidence level (e.g., 99%) would provide more reliability but less precision, as it would produce a wider confidence interval. Conversely, a lower confidence level (e.g., 90%) would produce a narrower confidence interval but offer less reliability. The 95% confidence level is a commonly accepted balance between these two factors.  In the context of our deep neural network (DNN) model for predicting lung adenocarcinoma (LUAD) prognosis, the confidence level helps to quantify the uncertainty associated with our predictions. It can offer clinicians and patients a more nuanced understanding of the prognostic predictions provided by our model.

Point 5. The authors need to compare their findings with recent cuproptosis-related gene signatures reported in the literature.

https://www.sciencedirect.com/science/article/abs/pii/S0010482523002962

https://www.sciencedirect.com/science/article/pii/S2405844023012987

Response 5: Thank you for your valuable feedback. We have compared the cuproptosis-associated genes signatures reported in the literature you mentioned and provided a detailed discussion and explanation in the discussion section. We found that although our study and these literature share commonalities in the investigation of copper death-related genes, the study subjects, analytical methods, and results differ.

Point 6. The authors will need to show more functional experiments to establish the role of genes in immune therapy and test its therapeutic potential.

Response 6: Thank you for your valuable feedback and suggestion to perform additional functional experiments to establish the role of genes in immune therapy and assess their therapeutic potential.  We appreciate the importance of conducting functional experiments to provide a deeper understanding of the mechanisms and potential therapeutic applications. However, due to the limited scope of our current study and the resources available, we are unable to conduct further experiments at this stage.  Nonetheless, we believe that our current findings provide a solid foundation for future research, which can delve deeper into the functional aspects and therapeutic potential of these genes in immune therapy. We hope that our work can serve as a starting point for other researchers in the field to expand upon and further validate our results through experimental approaches. In addition, we will further explore the roles of these genes in immunotherapy and test their therapeutic potential in our future research. Thank you again for your suggestions and support.

Point 7. Expand the logical validation of the experiments with a clear discussion.

Response 7: Thank you for your valuable suggestion. We agree on the importance of logical validation in experiments. Therefore, we have conducted a series of experimental validations on the potential therapeutic effects of BMS-754807. First, we screened 251 anti-tumor drugs and found that BMS-754807 showed significant sensitivity, which provided a theoretical basis for our subsequent research. Then, we conducted multiple concentration and time point experiments using the CCK-8 assay on the lung adenocarcinoma A549 cell line. The results showed that BMS-754807 could inhibit the proliferation of the A549 cell line at different concentrations and the inhibitory effect became more significant with increasing concentrations. In addition, we also observed cell migration and found that the number of migrating cells significantly decreased under the action of BMS-754807, further confirming its anti-tumor effect. Therefore, our experiments have validated the good anti-lung adenocarcinoma activity of BMS-754807, which provides experimental evidence for the performance of our deep neural network (DNN) model in drug screening.

Point 8. Authors should improve the figures as they are unreadable, small, and congested. It is recommended to use vector graphics images.

Response 8: Thank you for your valuable feedback. We have made modifications to the figures in the paper and improved them by using clearer vector graphics.

Point 9. How did the author determine each type of immune cell in the database?

Response 9: Thank you for your valuable comments. In this study, we used a method called single-sample gene set enrichment analysis (ssGSEA) to assess the immune microenvironment, including 16 types of immune cells and 13 pathway activities. ssGSEA is a computational method that estimates the enrichment level of predefined gene sets (such as a specific type of immune cell or a gene set of a certain pathway) in a sample based on its gene expression data. The advantage of this method is that it does not require a control sample, making it particularly suitable for our situation where we only have tumor samples without normal tissue samples. The 16 types of immune cell types and 13 pathway activities we used are defined and widely accepted in previous studies. These gene sets can be found in public databases such as the Molecular Signatures Database (MSigDB). In our study, we calculated the enrichment scores of these predefined gene sets in each sample, which can be viewed as the relative abundance or activity of that type of immune cell or pathway activity in the sample. In this way, we were able to obtain an estimate of the abundance of each type of immune cell in each sample.

Point 10. How was the optimal thresholds determined? Clearly clarify this point wherever it has been used.

Response 10: We greatly appreciate your suggestion and have explicitly described the principles for determining these thresholds in the Methods section of our paper. In our study, there are two places where we needed to determine the optimal thresholds: 1) in the deep neural network (DNN) model to determine the risk score threshold for distinguishing high-risk and low-risk patient groups; 2) in the analysis of tumor mutation burden (TMB) to determine the thresholds for high and low TMB groups. For the risk score threshold in the DNN model, we used receiver operating characteristic (ROC) analysis and calculated the area under the curve (AUC) to determine the optimal threshold. Specifically, we chose the threshold that maximized both sensitivity and specificity. In the TMB analysis, we divided the samples into high and low TMB groups based on their TMB scores. Specifically, we chose the threshold that maximized the difference in survival curves between the two groups (according to the log-rank test of Kaplan-Meier survival analysis) as the optimal threshold.

Point 11. The section and sub-section numbering is missing.

Response 11: Thank you for your valuable suggestion. We have added chapter and subsection numbering in the manuscript as per your suggestion.

Point 12. Captions/titles of the figures should be improved. For e.g. Figure 1 title can be made concise.

Response 12: Thank you for your valuable feedback. We have improved the titles of the figures and tables in the manuscript.

Point 13. The legends, axes titles in the figures should be stated clearly.

Response 13: Thank you for your valuable suggestions. We have made improvements to the manuscript according to your comments.

Point 14. Include a separate section for the related work.

Response 14: Thank you very much for your valuable suggestion. Our relevant work is presented in the introduction section of this article, in order to better introduce the background of our research and its connection to existing studies. This is also in accordance with the requirements and regulations of the journal in which we have published our research. Thank you once again for your suggestions and support.

Point 15. Include a separate section for the conclusions.

Response 15: Thank you for your valuable feedback. We have now made the conclusion section a separate chapter as suggested.

Round 2

Reviewer 1 Report

The authors have taken into account some of the revisions indicated, objectively improving the text.

However, I still think that the DNN model used is very basic and, above all, it is not explained why this model was chosen and not another. I think that not only the model used and its parameters should be explained, but also why that model was chosen and why those parameters are used. Including a figure of the model where L1 and L2 are explained.

Author Response

Point 1: However, I still think that the DNN model used is very basic and, above all, it is not explained why this model was chosen and not another. I think that not only the model used and its parameters should be explained, but also why that model was chosen and why those parameters are used. Including a figure of the model where L1 and L2 are explained.

 Response 1:

Thank you for your insightful comments. We agree that our original manuscript could benefit from a more detailed explanation of our model choice and its comparison with traditional models.

 The primary reason we chose a Deep Neural Network (DNN) model over the traditional Cox proportional hazards model is due to the DNN's ability to capture complex, non-linear relationships between predictors and the response variable, which in this case is patient survival time. This is especially relevant for high-dimensional genomic data where complex interactions are expected.

 The Cox proportional hazards model, on the other hand, assumes a linear relationship between the log-hazard and the predictors, and that the hazard ratios are constant over time (proportional hazards assumption). While the Cox model has been widely used in survival analysis due to its interpretability and statistical properties, it may not fully capture the intricate relationships in genomic data, which could potentially lead to a loss of predictive accuracy.

 In our study, we empirically compared the DNN model with the Cox model using the Area Under the Curve (AUC) of the Receiver Operating Characteristic (ROC) at different time points. The results demonstrated the superior performance of the DNN model at all time points, supporting our choice of the DNN model.

 We understand that the choice of model should depend on both the nature of the data and the specific objectives of the analysis. We believe that the DNN model is an appropriate choice for our study, given its ability to handle high-dimensional genomic data and its superior empirical performance. However, we acknowledge that other types of models may also be effective and encourage further studies to explore these possibilities.

 In the revised manuscript, we have provided a more detailed justification for our choice of the DNN model and its comparison with the Cox model. We have also added a new figure to visually demonstrate the architecture of our DNN model and the application of L1 and L2 regularization. We hope that these changes will address your concerns. Thank you again for your helpful suggestions.

Reviewer 3 Report

Authors have addressed most of the comments, successfully. Kindly improve the quality of the figures. 

Minor English language editing is required.

Author Response

Point 1: Authors have addressed most of the comments, successfully. Kindly improve the quality of the figures.

Response 1: We sincerely appreciate your valuable comments and suggestions. Following your advice, we have made concerted efforts to enhance the quality of our figures. Specifically, we have worked on improving their resolution and clarity to make them more reader-friendly. We have also ensured that all the labels and legends are clear and accurately represent the data in the figures.  We believe these changes have significantly improved the visual representation of our work and hope they meet your expectations. Once again, thank you for your constructive feedback. We look forward to hearing from you. 

Round 3

Reviewer 1 Report

I thank the authors for taking into consideration the changes requested in the last revision. With the DNN outline used I believe the article is now clearer and can be published.